# Suspected Permethrin-Containing Powder Bath Poisoning in a Flock of Mountain Quail (*Oreortyx pictus*)

**DOI:** 10.3390/ani15101428

**Published:** 2025-05-15

**Authors:** János Gál, Miklós Marosán, Míra Mándoki, Lilla Dénes, Miklós Süth, Dániel Pleva, József Lehel

**Affiliations:** 1Department of Exotic Animal-, Wildlife-, Fish- and Honeybee Medicine, University of Veterinary Medicine Budapest, István u. 2., 1078 Budapest, Hungary; gal.janos@univet.hu (J.G.); marosan.miklos@univet.hu (M.M.); 2Department of Pathology, University of Veterinary Medicine Budapest, István u. 2., 1078 Budapest, Hungary; mandoki.mira@univet.hu (M.M.); denes.lilla@univet.hu (L.D.); 3Department of Food Hygiene, Institute of Food Chain Science, University of Veterinary Medicine Budapest, István u. 2., 1078 Budapest, Hungary; suth.miklos@univet.hu; 4National Laboratory for Infectious Animal Diseases, Antimicrobial Resistance, Veterinary Public Health and Food Chain Safety, University of Veterinary Medicine Budapest, István u. 2., 1078 Budapest, Hungary

**Keywords:** permethrin poisoning, mountain quail, *Oreortyx pictus*, clinical chemistry, hematology, pathological changes

## Abstract

The prevention or treatment of external parasites is very important to maintain the health of livestock and can reduce the potential for economic loss. However, antiparasitic drugs, such as pyrethroids, can enter the body and induce poisoning. This study investigated the symptoms and pathological changes in permethrin poisoning in mountain quails, in order to improve understanding of the process and guide medical treatment.

## 1. Introduction

The prevention or treatment of parasitic diseases is vital in reducing economic losses caused by health problems and animal welfare in livestock. To this end, various ectoparasitic substances and products are available on the market, including organochlorines, organophosphates, pyrethrins and pyrethroids, and macrocyclic lactones. Of those, pyrethroids are recommended against a variety of ectoparasites such as flies, fleas, mite, lice, and ticks on companion and livestock animals in settings such homes, gardens, and farms [1]. Permethrin is also recommended to use against ectoparasitic infestations in several wild bird species, such as purple martins (*Progne subis*) [2], cliff swallows (*Hirundo pyrrhonota*) [3], barn swallows (*Hirundo rustica*) [4], blue tits (*Cyanistes caeruleus*) [5,6], pied flycatchers (*Ficedula hypoleuca*) [6], red grouse (*Lagopus lagopus*) [7], tropical mockingbirds (*Mimus gilvus*) [8], black-faced grassquits (*Tiaris bicolor*) [8], and barn owls (*Tyto alba*) [9].

All pyrethrins and pyrethroids have insecticidal and repellent properties. They have been used since the 1900s as plant-protecting pesticides, as insect repellents, and in the prevention and treatment of diseases caused by ectoparasitic insects in the animal health [10]. Permethrin, first marketed in 1977, is a cyclopropane carboxylate ester derivative. Currently, more than 3500 authorized insecticidal products containing permethrin are marketed worldwide [11]. Veterinary medicinal products containing different types of pyrethroids, in various concentrations, are authorized as different preparations including solid (e.g., powder, ear tags) and liquid forms (e.g., dip solution, spot-on, spray, shampoo).

Pyrethrins are derived from the flowers of Chrysanthemum plant species (e.g., *Chrysanthemum cinerariaefolium*, *Chrysanthemum cineum*). They can be found in the pollen as different pyrethrolone, cinerolone, and jasmolone esters of chrysanthemic acid or pyrethric acid (e.g., pyrethrin I and II, cinerin I and II, and jasmolin I and II) [12,13]. However, pyrethrins are highly unsaturated, easily oxidized compounds that quickly become ineffective when exposed to air. As a result, they are not frequently used in plant and animal health, but synthetic derivatives of pyrethrins, termed pyrethroids, have been developed, which eliminate these problems and are more stable in contact with light, oxygen, and moisture [14]. Pyrethroids are classified into two groups: type I pyrethroids (without alpha-cyano group in the molecule), e.g., allethrin, resmethrin, tetramethrin, bifenthrin, and permethrin, and type II pyrethroids (containing alpha-cyano group in the structure), e.g., deltamethrin, cyhalothrin, cypermethrin, cyfluthrin, fluvanilate, and fenvalerate [15,16].

Permethrin induces pronounced muscle and nerve disorders on insects (including useful insects such as honeybees and other pollinators) and worms, and is highly toxic to poikilotherms such as fish but also other aquatic organisms [1,17,18,19,20,21]. However, mammals are less sensitive (excepting cats) [22,23,24], and it is practically non-toxic in birds [25,26,27].

Pyrethroids induce hyperexcitability by disturbing membrane depolarization, inhibiting the closure of channels in the membrane (sodium, chloride, calcium, gamma-aminobutyric acid). The voltage-gated sodium channels are particularly susceptible, causing muscle tremors, twitching, fasciculation, or even convulsion due to repetitive neuron cell response given to a single stimulus [28,29,30]. The onset of toxicosis caused by pyrethrins and pyrethroids can manifest within seconds (knockdown) in susceptible animal species, and the duration of the effects can vary. Type II pyrethroids have a longer-lasting effect on neuron cell membranes than type I derivatives [31].

Though there is limited pharmacokinetic information on permethrin in birds, the pharmacokinetic profile of pyrethroids (including, e.g., cypermethrin, fluvalerate, and deltamethrin) in different mammals (e.g., rats, mice, humans), and birds (Japanese quails, bobwhite quails, Leghorn laying hens) is summarized based on the current literature. Generally, the absorption rate of pyrethroids and their penetration into the circulation after dermal application is relatively low (only about 2%) [32,33]. However, it is higher after ingestion (40–60%) or inhalation [15,34] depending on the substituents of the product or the mode of application [35]. After absorption pyrethroids are distributed in the body reaching relatively higher concentration in the fat tissue, liver, kidneys, and the nervous system. They are rapidly metabolized by ester hydrolysis and/or cytochrome P450-dependent monooxygenases to inactive metabolites which are excreted primarily in the urine. In mammals, 90% of the dose is excreted in urine compared to 40–90% in the excreta of birds, depending on the type of pyrethroids [23,31,35,36].

In case of pyrethroid poisoning, which generally manifests in acute form in more sensitive species, such as felines and poikilotherms, the following clinical signs and symptoms can be observed: excessive salivation, vomiting, muscle tremors, seizures, dermal reactions such as excessive itchiness, tachypnoea, and ataxia. However, death is rare in mammals, but it may occur in susceptible species [7,8,9,10,11,12,13,14,15].

Toxicosis caused by pyrethroids can be manifested even within seconds (knockdown) in the susceptible animal species, and their effects also last for a short time [7,8,9,10,11,12,13,14,15].

Permethrin is described as effectively nontoxic to birds, as generally, the acute oral LD_50_ is more than 1 g/kg [15,36]. For domestic birds, the oral LD_50_ value is thought to be 7–32 g/kg, specifically 11.3 g/kg for mallard ducks (*Anas platyrhynchos*), and 13.5 g/kg in Japanese quails (*Coturnix japonica*) [37,38,39,40]. However, toxic data are not available for mountain quails. The oral median lethal dose (LD_50_) value of permethrin was between 383 mg/kg and >2 g/kg in rats and was between 214 mg/kg and >5 g/kg in mice, respectively. The permethrin is less toxic dermally resulted in higher subcutaneous LD_50_ such as 6.6 g/kg in rats, >10 g/kg in mice, and 2 g/kg in rabbits [37,41].

Commercial insecticidal products can also contain piperonyl butoxide, which has a synergistic property on the effect of pyrethroids in insects, by binding to cytochrome P450 enzymes and inhibiting detoxification [16,42]. Piperonyl butoxide has low toxicity in mammals. Its acute oral LD_50_ is 4570–12,800 mg/kg in rats, and 2700–5300 mg/kg in rabbits. After dermal application, the LD_50_ is 7950 mg/kg in rats, and >2000 mg/kg in rabbits. The acute inhalational LC_50_ is >5900 mg/L in rats [43,44]. The exact data of the toxicity for piperonyl butoxide in birds are not available in the current literature.

The antiparasitic product used for to treat a flock of mountain quail (*Oreortyx pictus*) in this study also contained geraniol. Geraniol (3,7-dimethylocta-trans-2,6-dien-1-ol) and its derivative acyclic monoterpene alcohols (C_10_H_18_O) occur naturally as essential oils in plants, particularly in *Pelargonium* spp. such as *P. odoratissimum*, and other *Pelargonium* spp. They have potential effects against pests due to their insecticidal, acaricidal, and larvicidal activity against some worms such as roundworms *Contracaecum* species and nematodes, e.g., *Anisakis* spp. Furthermore, they have repellent and/or antifeedant properties [45,46,47,48]. The toxic potential of geraniol has been examined in various animal species, including carcinogenicity and mutagenicity tests. It has been shown to have low mutagenic potential [49]. It has low toxicity in mammals. Its LD_50_ values were found to be 3.6 g/kg orally in rats, 1.09 g/kg subcutaneously in mice, 4 g/kg intramuscularly in mice, and 5 g/kg dermally in rabbits. Due to the long-term application of geraniol fed to rats at a dose of 10,000 mg/kg for 16 weeks and 1000 mg/kg for 28 weeks, adverse effects were not observed [37,49]. Toxicity data of geraniol are not available for birds. There may be synergistic effects of geraniol with combination of antimicrobial agents, but this effect is not documented in case of antiparasitic chemicals used concomitantly [49].

The aim of the study was to describe unwanted pyrethrin poisoning in mountain quails (*Oreortyx pictus*) including the potential source, diagnosis, differential diagnosis, and giving the potential treatment principles.

## 2. Materials and Methods

### 2.1. Animals

The study was performed on an exotic game bird farm specialized for housing and breeding of 30 different quail and partridge species in Hungary. From those, seven young, 5-month-old (four males, three females) nest mate mountain quails weighing 382–475 g were kept together in an aviary. While the inclusion of control groups is essential in experimental toxicological studies, it is not feasible in the context of an unplanned, real-life poisoning case. Therefore, the absence of a control group should be considered a practical limitation rather than a methodological flaw.

### 2.2. Housing and Nutrition

The birds were housed in an outdoor aviary (width: 1.5 m; length: 2.5 m; height: 1.8 m) with a grid bench floor partially covered with wooden planks. Rest boards were placed on the floor, with 40% of the floor providing a solid walking surface. The animals were fed from a self-feeder ad libitum, with a mixture of seed including millet (40%), foxtail (20%), wheat (20%), canary grass (5%), flax (3%), and rape (2%). Their feed was supplemented by a small quantity of apples and chickweed twice a week. Fresh drinking water was always available.

### 2.3. Treatment

Against different ectoparasitic insects, the environment of the seven mountain quails were treated with Piret-Kill insecticidal powder (Metatox Pesticide Manufacturer and Distributor Ltd., Szeghalom, Hungary), which is recommended against cockroaches, ants, fleas, bed bugs, and other concealed insects. It contains 0.5% permethrin, 0.15% extract of *Chrysanthemum cinerariae* folium, 0.3% piperonyl butoxide, and 0.01% geraniol, which is authorized as a biocidal preparation based on the legal regulation under the Main Group 3 (pest control), Product-type 18 (insecticides, acaricides, and products to control other arthropods) [50]. The product was mixed with fine-grained sand (ratio 1:5), and placed in a powder bath in the aviary in autumn, as per the product instructions.

### 2.4. Clinical Chemistry and Hematological Analysis

A blood sample was taken from one of the quails showing clinical signs, and a hematological analysis was performed at the Department of Clinical Pathology and Oncology, University of Veterinary Medicine Budapest. The clinical chemistry and hematological analysis were performed by the Beckman Coulter Olympus AU400 and AU480 automatic machines (Beckman Coulter/Olympus, New York, NY, USA), and the detailed data for each analyte (manufacturer, reagent identifier) are given in Table 1.

### 2.5. Necropsy and Sampling

Necropsy was performed on all seven dead birds. Samples were taken from organs which showed alterations, such as liver and kidney. They were placed into a 10% buffered formaldehyde solution to fix them and then embedded into paraffin. Sections 2–3 μm thick were made, and after deparaffinating on slides, they were stained with hematoxylin eosin. Furthermore, liver samples were taken for Oil-Red-O staining. After freezing without fixing, sections of 3–5 μm thick tissue were drawn on a slide. Oil-Red-O dye was dissolved in propylene glycol, and the slides were stained for 5 min. Then, they were stained with alum hematoxylin for a few seconds followed by rinsing and covered by glycerin-gelatin. Liver samples were also taken for microbiological investigation. The samples were inoculated onto blood agar (Blood agar (base), Sigma-Aldrich, Merck Life Science Ltd., Budapest, Hungary) and Drigalski culture media (Drigalski agar, Biolab Diagnostics Laboratory Inc., Budapest, Hungary), which was incubated for 24 h at 37 °C in an aerobic atmosphere. In addition, liver samples were taken for a PCR analysis during the necropsy to determine the potential presence of Avi-adenovirus (including all types of it), avian hepatitis E virus, and circovirus (beak and feather disease).

Samples were not taken from the stomach contents, the organs, and other biological items for toxicological analysis.

### 2.6. Nucleic Acid Extraction and PCR Analysis

Viral RNA and DNA was extracted from liver via the MagCore Viral Nucleic Acid Kit (RBC Bioscience Corp., New Taipei City, Taiwan) using the MagCore Plus II Automated Nucleic Acid Extractor (RBC Bioscience Corp., Taiwan), according to the manufacturer’s instructions. The nucleic acids extracted from the samples were removed from 60 μL elution buffer (EB) (RBC Bioscience Corp., Taiwan) for each sample and then immediately subjected to PCR analysis or stored at −80 °C until further examination.

The presence of Avian Hepatitis E virus (HEV) was analyzed by in-house reverse transcription-polymerase chain reaction (RT-PCR) using QIAGEN OneStep RT-PCR kit (QIAGEN, Hilden, Germany) with the following heat profile: 50 °C for 30 min, 95 °C for 15 min, 40 cycles at 95 °C for 30 s, at 58 °C for 30 s, and at 72 °C for 1 min, and then final elongation at 72 °C for 3 min. The mixture contained 2 µL 5× buffer, 0.4 µL dNTP (10 mM), 0.4 µL Enzyme Mix, 0.1 µL RNase inhibitor (ThermoFisher Scientific, Vilnius, Lithuania), 5.7 µL RNase free water, and 0.75 µM end concentration of both primers (HEV F: 5′-GGTTCGTGAACATTGGAGAC-3′ and HEV R: 5′-CAGGCACAACAGGTGTAACT-3′) designed to amplify a ~250 bp long region of the capsid protein coding gene and a 1.5 µL template.

The Avi-adenovirus sp. (previously Fowl adenoviruses) detection was performed using the Hexon1: 5′-TGGACATGGGGGCGACCTA-3′ and Hexon2: 5′-AAGGGATTGACGTTGTCCA-3′ primers targeting a conserved ~1219 bp long region of the hexon gene with the following PCR program: 94 °C for 3 min, 35 cycles at 94 °C for 30 s, 60 °C for 30 s, and 72 °C for 1 min, with a final extension cycle at 72 °C for 10 min [51,52].

The detection of Circovirus parrot (previously Beak and feather disease virus) responsible for psittacine beak and feather disease was performed using the BFDV F (5′-AACCCTACAGAC GGCGAG-3′) and BFDV R (5′-GTCACAGTCCTCCTTGTACC-3′) primers targeting the open reading frame 1 (ORF1) of the BFDV genome, amplifying a 717 bp long fragment. The PCR profile was the following: 94 °C for 7 min, 32 cycles at 94 °C for 1 min, 60 °C for 30 s and 72 °C for 1 min, and then 72 °C for 10 min [53].

The detection of *Avi-adenovirus* sp. and Circovirus parrot was performed using the QIAGEN AllTaq PCR Master Mix kit (QIAGEN, Hilden, Germany). The end volume of the PCR mix was 20 μL and contained 5 μL 4× AllTaq MM, 0.1 µ MM Tracer, 11 μL RNase free water, 0.25 μM end concentration of the forward and reverse primers, and 2.5 μL of the template. Every PCR reaction was run in a QIAamplifier 96 machine (QIAGEN, Hilden, Germany).

## 3. Results

### 3.1. Clinical Signs

When the clinical signs manifested, the powder bath tray was removed from the aviary. The birds became noticeably less active three days after starting of the treatment (day 0). Several quails were depressed and sitting in one place with ruffled plumage. Other characteristic neurological signs of pyrethrin toxicosis (ataxia, tremor, seizures) were not observed. Six birds out of seven died on day 4 after the beginning of the treatment, and the remaining bird died in the morning on the next day.

On the exotic game bird farm, the different bird species and their different groups were housed in several outdoor aviaries in the same area. However, clinical signs of poisoning and other diseases were not observed on the other birds.

### 3.2. Hematology

The results of the hematological analysis from one of the affected birds are presented in Table 2. Certainly, the hematological analytical results of this one bird may not be representative of the entire affected group, but it can give information regarding the potential hematological changes due to permethrin poisoning.

### 3.3. Necropsy

The necropsy revealed that the birds had suffered damage to the liver and kidneys, whilst other organs appeared unharmed. In the opened coelomitis, the placement of organs was normal. The liver of the necropsied bird was diffusely mottled, friable, and mildly enlarged, according to Figure 1A.

The gallbladder was dark green in color, elongated, taut, and lying on the visceral surface of the liver. The bile was dark green, thinly flowing, while the gallbladder wall was extremely thin. In the posterior part of the body cavity, in the fat deposit behind the gizzard, grayish-white areas with a waxy sheen were visible, which differed from the usual fat bodies in their rigidity (Figure 1B).

The kidneys were mottled and bilaterally enlarged, according to Figure 1C. The veins returning to the kidneys were filled with blood.

### 3.4. Microbiological Test

The results of liver samples tested for bacteria in an aerobic atmosphere on blood agar and selective Drigalsky medium were negative.

### 3.5. PCR Test

Similarly, the findings of a PCR analysis of liver samples to detect Avi-adenovirus, avian hepatitis virus, and circovirus (beak and feather disease) were negative.

### 3.6. Histopathology

The hepatic parenchymal structure was disrupted, and many cells at different stages of degeneration and apoptosis were observed. Only a few cells with prominent nucleoli showed perichromasy. Most hepatocytes were rounded, and some had nuclei in the lysis phase; others were in a coreless state, homogeneously stained with eosin, suggesting complete cell death and the presence of eosinophilic cytoplasmic inclusions. In addition, similar nuclear inclusion associated with hepatitis has been frequently seen in poultry due to adenovirus infection. However, the presence of adenovirus was not detected by PCR method. Similarly, cell death together with degeneration of the cell nucleus may develop in the liver because of toxicoses which can induce similar intranuclear lesions. In addition to solitary cell degeneration, foci of micro-necrosis resulting from so-called group-coordinated liver cell destruction were also observed without any significant inflammatory cell infiltration. Signs of heterogeneous vacuolization were evident in the cytoplasm of cells, still functioning but already showing signs of decline and shifting towards degradation (Figure 2A).

In liver sections stained with Oil-Red-O, high heterogeneity was seen in the fat content, distribution, and localization of the cytoplasm in individual cells. In cells that already showed signs of regression, large heterogeneous droplets of fat accumulated, while liver cells with nuclei contained fine, tiny droplets, evenly dispersed in the cytoplasm, but in greater amounts than physiological (Figure 2B).

The tubular structure of the kidney was segmentally destroyed, the entire epithelial cell population disintegrated and died, and the tubule cavity was filled with cell debris (Figure 2C). Furthermore, the dilatation and abundance of intertubular blood vessels were seen with extravasation of red blood cells.

## 4. Discussion

Permethrin toxicosis was strongly suspected in these birds based on the circumstances of exposure and the ruling out of other potential causes of the birds’ clinical signs. This is the first case of pyrethroid poisoning in birds reported in Hungary. Generally, birds are less sensitive to pyrethroids, and thus, cases of poisoning are rarely described in the scientific literature.

During the poisoning, acute malfunctioning of liver was observed with regressive lesions of the liver cells, solitary and necrotic destruction, and the disturbance of fatty metabolism in the cytoplasm accompanied by multiplex liponecrosis.

Based on the current literature, generally, pyrethroids, and therefore permethrin, can induce liver failure [13,15]. However, Bulgarella et al. found that the application of 1% permethrin solution to zebra finches did not induce hepatotoxic effects (*Taeniopygia guttata*) [25,55].

In addition to the affected liver, the segmental destruction of the renal tubule epithelium was observed. The poisoned birds also showed a high degree of degeneration of epithelial cells. Similar pathological changes have been described in rats; permethrin (75 mg/kg b.w./day) induced swelling and hyperchromatosis in the nuclei of the hepatic parenchymal cells, and swollen proximal tubules were detected in the kidneys [56].

The severe impairment and loss of both liver and kidney function also produced characteristic changes noted during clinical chemistry and hematological analysis. Elevated activities of AST and ALKP were recorded. AST, which indicates liver damage, was three times higher than usual, and GLDH nearly ten times higher. Moreover, the total bile acid levels of liver cell specific were significantly higher than expected. In this case, liver-specific bile acid, characterizing the excretory function of the organ, had tripled. In addition to extremely high uric acid levels in serum, high levels of bile acid are a sign of loss of kidney function. Above five times the normal value, an increase in the level of ions (sodium, chloride), including hyperkalemia, is likely due to significant cell death.

Piperonyl butoxide has synergistic effect in combination with pyrethroids with low toxicity in mammals; however, data are not available in birds. The geraniol may have synergistic effects with antimicrobials, but it is not described with ectoparasitic agents [16,42,45].

Simultaneous liver and kidney failure may be manifested due to feed toxicosis (e.g., mycotoxins, high peroxide levels). However, all the other birds kept on the farm ingested the same feed without any clinical signs. Furthermore, the other potential causative agents or microorganisms (e.g., fowl adenovirus mimicking “inclusion body hepatitis” noted in broiler chicken worldwide) causing liver damage have been excluded by a hematological analysis of the blood and different microbiological and other tests described previously. Thus, we feel able to suggest a causal relationship with the uptake of permethrin.

Despite compliance with the instructions and regulations, in this case, the antiparasitic treatment proved fatal. In cases such as these, birds can take up the ectoparasitic agents in such quantity that it can cause severe toxicosis. In this case, presumably the birds did not receive enough crushed grit and pecked at the treated sand in the dust bath to counter this. Based on the toxicity profile and properties of geraniol and piperonyl butoxide, they could not have contributed to the development of poisoning, but they and other potential contributing factors cannot really be ruled out. However, during the necropsy, no excessive materials were found in the gastrointestinal tract besides seeds and a few pebbles. Thus, the possible oral and/or inhalation exposure due to normal dust bathing and subsequent preening behaviors could have also influenced the outcome of the poisoning. The active substance absorbed from the gastrointestinal or the respiratory tract due to the pyrethrin-coated sand led to severe liver and kidney damage, causing the birds to die due to organ failure.

## 5. Conclusions

We would like to draw attention to the potential toxic effects of pyrethroids. Though previously considered toxic only to poikilotherm animals, and moderately or non-toxic in warm-blooded species, if not used appropriately these preparations can pose a risk of poisoning. It is important to use ectoparasitic agents that are considered less toxic, and they must be used based on label instructions. Basically, the oral exposure by the consumption of the contaminated sand from the powder bath during the normal dust bathing and subsequent preening behavior could have induced the poisoning, although inhalation could have contributed to it.

The prudent use of ectoparasitic preparations is recommended. However, if poisoning is suspected, the potentially poisonous source should be immediately withdrawn (in our case, dust baths treated with a permethrin-containing product) and treatment supporting liver and kidney function should be started immediately. Due to the fatty disorder of the liver, in addition to giving amino-acid-containing infusion supporting liver function, we also recommend vitamin therapy including fat- and water-soluble vitamin-containing products, added to drinking water. There are positive reports of the administration of organic acids, e.g., acetic acid, in a volume of 1 mL/L drinking water which directly supports liver function in cases of hepatopathy [57].

## Figures and Tables

**Figure 1 animals-15-01428-f001:**
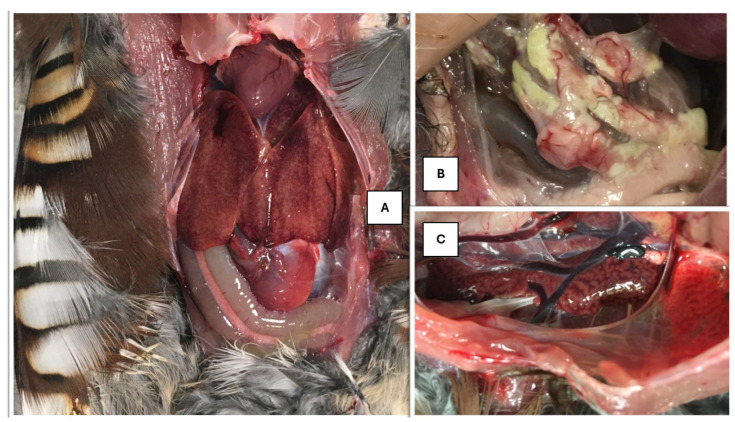
(**A**) Mottled liver. (**B**) Waxy areas at the fatty body in the abdominal cavity. (**C**) Enlarged, marbled kidney.

**Figure 2 animals-15-01428-f002:**
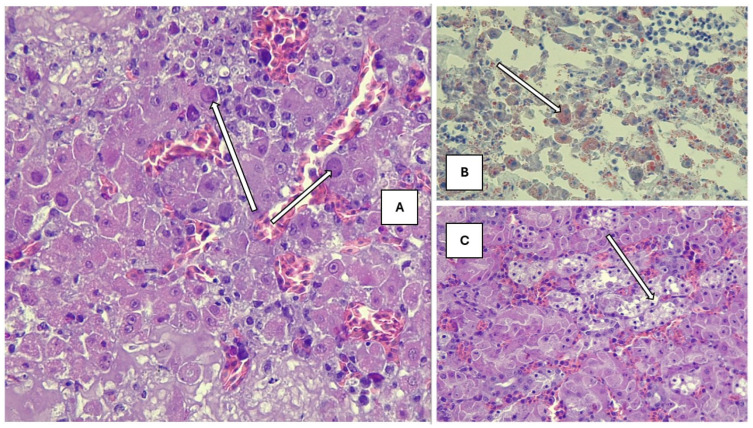
(**A**) Affected liver with eosinophilic cytoplasmic inclusions (H-E, 200× magnification). (**B**) Pathological fatty infiltration in the liver (Oil-Red-O, 200× magnification). (**C**) Tubular epithelial vacuolization (H-E, 200× magnification). Histopathological changes are pointed by arrows.

**Table 1 animals-15-01428-t001:** Summary of the blood parameters determined and the measurement methods.

Parameters	Measurement Methods
Hematocrit (%)	micro-hematocrit method, 10,000 g, 5 min
White blood cells count (×10^9^/L)	smear analysis
Thrombocyte count (×10^9^/L)	smear analysis
Heterophil granulocyte segmented (%)	smear analysis
Albumin (g/L)	Beckman Coulter, OSR6102
Globulin (g/L)	calculated value (TP-Alb)
Total protein (g/L)	Beckman Coulter, OSR6132
AST (U/L)	Beckman Coulter, OSR6109
ALT (U/L)	Beckman Coulter, OSR6107
ALKP (U/L)	Dialab, D95560
GGT (U/L)	Dialab, D95604
GLDH (U/L)	Diasys, 124119910021
Bile acid total (µmol/L)	Diasys, 122389910930
Total bilirubin (µmol/L)	Beckman Coulter, OSR6112
α-amylase (U/L)	Beckman Coulter, OSR6182
Lipase (U/L)	Dialab, D01440
Glucose (mmol/L)	Beckman Coulter, OSR6121
Triglyceride (mmol/L)	Beckman Coulter, OSR61118
Total cholesterol (mmol/L)	Beckman Coulter, OSR6116
Uric acid (µmol/L)	Dialab, D95459
Creatinine (µmol/L)	Dialab, D95595
Inorganic phosphate (mmol/L)	Beckman Coulter, OSR6122
Total calcium (mmol/L)	Dialab, D01376
Sodium (mmol/L)	Backman Coulter ISE
Potassium (mmol/L)	Backman Coulter ISE
Chloride (mmol/L)	Backman Coulter ISE
Iron (µmol/L)	Beckman Coulter, OSR6186
Magnesium (mmol/L)	Beckman Coulter, OSR6189
CK (U/L)	Beckman Coulter, OSR6179
LDH (U/L)	Beckman Coulter, OSR6126

**Table 2 animals-15-01428-t002:** Results of blood biochemistry of the longest-surviving seventh bird.

Parameter	Measured Value	Reference Value *
AST (U/L)	1542	432.1
ALT (U/L)	37	238.0
ALKP (U/L)	1156	829.69
GLDH (U/L)	38.7	3.91
total bile acid (µmol/L)	343.4	92.83
total bilirubin (µmol/L)	5.4	ND
triglyceride (mmol/L)	0.9	1.05
total cholesterol (mmol/L)	1.9	3.57
uric acid (µmol/L)	1006.8	293.82
phosphorous (mmol/L)	2.96	ND
total calcium (mmol/L)	1.7	2.79
potassium (mmol/L)	5	2.06
sodium (mmol/L)	162.8	156.42
chloride (mmol/L)	122.6	110.8
magnesium (mmol/L)	1.89	1.16
LDH (U/L)	17,265	473.92

ND = No data; * [54].

## Data Availability

The raw data supporting the conclusions of this article will be made available by the authors on request.

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
