# Peer review of "Suspected Permethrin-Containing Powder Bath Poisoning in a Flock of Mountain Quail (Oreortyx pictus)"

_animals, 2025, doi:10.3390/ani15101428_

Round 1

Reviewer 1 Report (Previous Reviewer 1)

Comments and Suggestions for Authors
  1. The new title about suspected toxicosis does not match the statement in the abstract that this is a report of a case of toxicosis.
  2. The new section about PCR analysis beginning with Line 184 seems excessive given that the results were all negative.
  3. The case data provided does not adequately support a diagnosis of permethrin toxicity.

Author Response

Reviewer 2 Report (Previous Reviewer 2)

Comments and Suggestions for Authors

This still needs some revision.

The abstract needs more information. It does not contain any information on the number of birds affected or the fact that exposure was fatal in all cases. The reader has no information on the scale of the incident. It also states that permethrin toxicosis HAD occurred when the title is SUSPECTED permethrin poisoning.

Generally, birds are less sensitive to pyrethrin and pyrethroid toxicosis compared to mammals, however, suspected permethrin toxicosis was fatal in seven mountain quails (Oreortyx pictus) treated with a permethrin-containing powder bath. Signs started 3 days after the bath was placed in their enclosure and were non-specific with reduced activity and ruffled feathers. Unusually, the characteristic signs of permethrin poisoning were not observed, and infective causes of the birds’ clinical signs were ruled out. Histopathological evaluation, however,  revealed malfunctioning of the liver and kidneys; the liver appeared mottled and degenerated, and the kidneys were enlarged, marbled and degenerated. Specific therapy and antidotes to pyrethrins are not available; the treatment is supportive, including hepatoprotective agents and vitamin therapy.

Line 73

What does expressed mean here?

Line 72

Fish are poikilotherms so this sentence does not make sense.

.. worms, and is highly toxic to poikilotherms such as fish but also other aquatic organisms.

Line 87

‘Smaller amount’ here does not make sense.

…penetration into the circulation after dermal application is relatively low (only about 2%).

Line 93-95

This does not make sense.

...which are excreted primarily in the urine. In mammals 90% of the dose is excreted in urine compared to 40-90% in the excreta of birds, depending on the type of pyrethroid [23,31,35-36]

Lines 101-02

This statement is not referenced. What is ‘them’? Pyrethroids?

Line 105

You state permethrin is effectively non-toxic to birds, but you are showing otherwise, so this should be reworded.

Permethrin is described as effectively non-toxic to birds…

Line 108

The LD50 values given here – are they dermal or subcutaneous as these are not the same route of exposure.

Line 118

...in this study ALSO contained geraniol.

Line 140

As the birds were not all male, would nest mates be better than nest brothers?

Line 156

The instructions for the product include its use as a dust bath for birds?

Line 182

It would be worth noting that there was no toxicological analysis of the dust bath, stomach contents or other biological samples. 

Discussion

I would change the first line to

Permethrin toxicosis was strongly suspected in these birds based on the circumstances of exposure and the ruling out of other potential causes of the birds’ clinical signs. This is the first case of pyrethroid poisoning in birds reported in Hungary.

Line 304

Enzymes activities are measured not levels. So, elevated activities of AST and ALKP…

Line 323

...a causal relationship with the uptake of permethrin.

Comments on the Quality of English Language

This still needs some work to tidy up the English.

Author Response

Reviewer 3 Report (New Reviewer)

Comments and Suggestions for Authors

The manuscript by Gal et al. (Suspected permethrin-powder bath poisoning in a flock of 2 
mountain quail (Oreortyx pictus) contain very intersting information. Here minor comments to further improve manuscript.

1-Ethical approval number must be provided in materials and methods

2. In the necropsy and sampling section, specify the target collected organs and provide a reference for histopathological sample preparation

3-In figure 2, replace Councilman bodies with eosinophilic cytoplasmic inclusions

2-In Figure 2c, replace tubulonephrosis and nephropathy with tubular epithelial vacuolisation.

3- In Materials and methods section, explain How oil red o stain performed

Author Response

Reviewer 4 Report (New Reviewer)

Comments and Suggestions for Authors

Dear Authors,

the topic of this manuscript is highly relevant, and you have done a very good job in highlighting important aspects. This work provides important insights for future medical interventions by studying the symptoms and pathological changes associated with permethrin toxicity in mountain quails. Although some methodological limitations are present, particularly the lack of direct toxicological confirmation and the reliance on nonspecific pathological findings, these are understandable and they can be addressed. I am confident that the manuscript will develop into an even more impactful and valuable research.

Lines 22-23: Instead of “...to better understand the process…treatment plan” change with “...to improve understanding of the process and guide medical treatment”.

Lines 31-32: Change “… the liver appeared mottled and degenerated, and the kidneys were enlarged, marbled and degenerated” with “the liver appeared mottled and degenerated, while the kidneys were enlarged and marbled, showing signs of degeneration”.

Lines 25–34 & 218–225: According to the description of those lines the diagnosis of permethrin toxicosis is based solely on clinical sings, pathological, and exclusion criteria. However, no chemical analysis (HPLC or GC-MS) has been reported to assess the dosage of permethrin or its metabolites in body tissues or fluids.

Lines 162–170 & 230–233: The clinical and hematological analyses were conducted only on one bird, the longest surviving subject. This bird may not be representative of the entire affected group. I suggest to the authors to acknowledge this restriction in the limit of the study.

Lines 152–160: The manuscript does not provide any estimates of the actual dose of permethrin absorbed by birds (this should be specified within the limitations of the study). Given the known low dermal absorption in birds, oral or inhalation exposure likely played a major role. I recommend adding a discussion about this dynamic including any citations.

Line 221: Change “after starting of the treatment” with “after the beginning of treatment”.

Lines 226–228: Although the authors state that other birds in the facility did not show symptoms, no control group was established to compare data from treated and untreated birds. Add this aspect within the limits of the study

Lines 257–285: The liver and kidney lesions are consistent with toxic damage, but are not pathognomonic for permethrin poisoning. Findings similar to those of viral hepatitis (adenovirus) were observed, but were excluded via PCR. I recommend the authors to discuss alternative diagnoses more carefully.

Lines 152–160: The product used (Piret-Kill) is not authorized for use directly on birds, especially outdoor. Its off-label use as a dusting agent in quail enclosures requires further justification and discussion of regulatory implications.

Lines 110–116 & 313–315: Since the product contains piperonyl butoxide and geraniol, which could potentially act synergistically, their specific contribution to toxicity in this species is still unclear. I suggest further investigation and discussion.

Lines 103–109 & 294–296: The statement that permethrin is “practically non-toxic” requires better contextualization, as toxicity can vary considerably among different bird species. Species-specific variability in sensitivity is well documented and the discussion would benefit from referencing related species or by reporting the lack of data on mountain quail.

Line 344: Change “however, …” with “although, inhalation could also have contributed”.

Round 2

Reviewer 3 Report (New Reviewer)

Comments and Suggestions for Authors

Authors respond to all comments 

Reviewer 4 Report (New Reviewer)

Comments and Suggestions for Authors

Dear Authors,

the revisions made by you address several of the previously raised concerns and contribute to a more coherent presentation of the findings.

Best Regards

This manuscript is a resubmission of an earlier submission. The following is a list of the peer review reports and author responses from that submission.

Round 1

Reviewer 1 Report

Comments and Suggestions for Authors

Thank you for a greatly improved manuscript, however the addition of Lines 108-117 about geraniol, another ingredient in the applied product in addition to permethrin, pyrethrin, and piperonyl, needs to be expanded for better understanding. How does this first mention of geraniol on page 3 relate to the previous information and the overall case report, especially with the lack of detail provided compared to what is included for pyrethrins/pyrethroids and piperonyl?

Since the introduction states that pyrethrin/pyrethroid toxicosis is not expected in birds, the Discussion section does not adequately address the unexpected outcome in this case, including the potential contributions of the geraniol and the synergistic actions of the piperonyl. New Lines 273-278 presume excessive ingestion of the product; however the necropsy findings do not describe any excessive materials in the gastrointestinal tract besides seeds and a few pebbles. Contributions of oral +/- inhalation exposure due to normal dust bathing and subsequent preening behaviors should also be addressed.

Comments on the Quality of English Language

Thank you for the revisions to this manuscript, however the quality of the English language could still use significant improvement to avoid confusion. For example, in Lines 80-82, is the intent really to say that absorption is "faster" or instead "higher" after ingestion since this statement is then followed by a percentage (40-60%)? Similarly, in Lines 94-97 the sentence begins with stating permethrin is "effectively nontoxic" to birds but then follows with a number of oral LD50s in the 7-32 mg/kg range. Lines 197-198 seem to be an incomplete sentence.

Reviewer 2 Report

Comments and Suggestions for Authors

This is much better ,and I have a clearer understanding of what happened. Still a few little queries to address.

Line 121. The birds were all males?  They were nest mates? How old were they? How much did they weigh?

Line 165-676. Six birds out of seven died on day 4 after the treatment and the remaining bird died shortly after.

When did the 7th bird die? Not on day 4? How long after exposure was day 4 of treatment?

Table 2. Results of clinical chemistry and clinical biochemistry of the blood of the surviving bird.

This bird did not survive. Perhaps better to says ‘Results of blood biochemistry of the seventh bird’.

Line 195. The adrenal… not he adrenal…

Line 266 …and an increase in potassium levels are also known to be signs of hyperkalemia (due to significant cell death).

An increase in potassium is hyperkalaemia, so this does no make sense. Do you mean

‘An increase in the level of ions (sodium, chloride), including hyperkalemia, is likely due to significant cell death.

Line 262. There are positive reports of the administration of organic acids, e.g. acetic acid, in a volume of 1 ml/l drinking water which directly supports liver function in cases of hepatopathy.

This should be referenced.

Comments on the Quality of English Language

Some minor changes are required to improve the English. 

Reviewer 3 Report

Comments and Suggestions for Authors

Dear authors,

Please consider the next major and minor editing, as follows

TITLE

L2-L3 The title could be improved. A potential new version to consider is "Suspected pyrethrin-powder bath poisoning in a flock of mountain quail (Oreortyx pictus)".

L21-L30 Abstract

-. The term toxicosis is used to characterize the pyrethrin exposure of affected birds in the Abstract. The term "toxicosis" must be replaced with "suspected poisoning". Toxicosis could be used when a toxic agent is detected and measured in a target organ. 

-. I would like to recommend to change from "toxic liver and kidney dystrophy" to "affected liver and kidneys" (L26). In L27, "liver and kidney dystrophy" could be replaced by "malfunctioning of liver and kidneys".

INTRODUCTION

-. L94 to L117 This paragraph could be relocated in the Discussion section.

-. The aim of this study must be included at the end of the Introduction section.

MATERIAL AND METHODS

-. The authors are recommended to clarify if the naming of the birds is "Mountain" or "mountain" quails. The scientific name should be included only once when is mentioned for the first time

-. L130-L131 Was the drinking water fresh and sanitized?

-. L133 and L139 "Piret-Kill" is mentioned as the product used for the pyrethrin-powder bath in the studied birds. Please include the concentration of pyrethrin in this product, and name of the facturer, country and city of origin.

-. L143 The authors mentioned that a blood sample was obtained from a one of the mountain quails with clinical signs. However, no clinical signs were described in the subsection "2.1 Animals" but present in the subsection "3.1. Clinical sings". From my view, both subsections could be unified in one. How many birds were studied? How many birds were alive and dead at the time of evaluation? If there were live birds, please include the euthanasia methodology for the purpose of postmortem examination.

-. L149 Please consider to unify both Tables 1 and 2 to facilitate a fluent reading.

-. L153 Was 8% de concentration of the formalin used to fix tissues for the histopathologic analysis? Routinely, 10% neutral formalin is used for fixation and histopathologic evaluation of affected tissues.

-. L156 and L157 The authors should include the commercial name, manufacturer, city and country of origin of the media culture for bacteriologic analysis. The term "atmosphere" is recommended to be used instead of "environment".

-. L158 and L159 The authors should be more specific with the targeted viruses searched here. Which of the adenovirus, avian hepatitis virus and circovirus were studied? Cited references must be included to support this molecular methodologies.

RESULTS

-. L164 to L167 Please rewrite this paragraph.

-. L173 to L179. This paragraph should be reorganized. Please follow these recommendations

. Organs without gross lesions should not be included on this paragraph, such as plumage, skin and body orifices, as mentioned by the authors.

. The term "body cavity" needs to be replaced by celomic or abdominal cavity.

. The liver of the necropsied bird was diffusely mottled, friable and mildly enlarged, according to Figure 1A.

. The pictured pancreas has a dark pink colour. Did you include pancreas on the histopathologic study?

. In Figure 1B, the creamy and yellowish multiple areas is compatible to caseous airsacculitis instead of degenerated abdominal fat tissue.

. Kidneys are mottled and bilaterally enlarged, according to Figure 1C.

. The term "degenerated" is redundant to characterize liver and kidneys.

. The description of the lack of abnormalities of the oropharyngeal cavity, upper respiratory tract, lungs, spleen, proventriculus, gizzard, intestines and gonads can be deleted.

-. L201 to L203 The results of the microbiological test should be simplified.

-. L204 to L207 It is not clear if both PCR and sequencing results of the tested viruses were negative.

-. L208 to L240 According to Wikipedia: "A Councilman body, also known as a Councilman hyaline body or apoptotic body, is an eosinophilic globule of apoptotic hepatocyte cell fragments. Ultimately, the fragments are taken up by macrophages or adjacent parenchymal cells". According to Figure 2A, the presence of intranuclear inclusion bodies and peripheral chromatin should be ruled out in this liver section. An special stain for the observation of inclusion bodis is recommended to be included on the histopathologic study of the liver

The histopathologic description should be shortened in order to be concise. Please consider that the tubular epithelial cells are easily affected by postmortem changes mimicking degeneration and necrotic changes. 

-. L223 to L226 Please include more details on the histopathologic description of Figures 2 A to C. Please clarify what microscopic finding (e.g. hepatocellular vs interstitial) is related to the "pathological fatty infiltration"? 

DISCUSSION

-. Please use "current literature" instead of "scientific literature".

CONCLUSION

-. Please avoid the use of "In conclusion,..." below the heading of "Conclusion". 

-. L286 Please change from "use prudent" to "prudent use".

-. The recommendation of commercial products, as it is mentioned in L291, should be avoided.

-. Please be explicit if you consider the overexposure by oral route as the main portal of entry of this ectoparasitic product.

ADDITIONAL COMMENT 1. The term "toxicosis" is used several times in this manuscript. In this report, "poisoning" should be used instead.

ADDITIONAL COMMENT 2. Did the authors ruled out other causes of hepatic and renal damage such as mycotoxicosis? If yes, please explain how you discarded this entity.

Reviewer 4 Report

Comments and Suggestions for Authors

Poisoning in a flock of quail by pyrethrin

This case report is really interesting, and deserves attention, but I have some concerns of over-reach in the conclusions.  I don’t think that you can say that the permethrin caused the acute lesions described based on some chronic studies in the literature, the lack of more typical clinical signs, and the fact that a multi-ingredient product was used.  Additionally, though you do explain the difference between pyrethrins, pyrethroids, and permethrin, you also seem to be using these terms interchangeably, for example, in the title you state it’s a pyrethrin product when it’s a permethrin product.

Title:  Title is incorrect.  It is a permethrin-containing powder.

Abstract:

P1, L21:  “Generally, birds are less sensitive” this sets up a comparison but doesn’t tell us to what you’re comparing birds.

Introduction:

P1, L35:  Parasitic diseases are also an animal welfare issue. 

P1, L43: Italics error.

P2, L60:  I had to read this passage a couple of times to achieve clarity.  Maybe “synthetic derivatives of pyrethrins, termed pyrethroids,”

P2, L66:  Permethrin is not similar to pyrethroids, it is a pyrethroid.

P2, L79-86:  The specific pyrethroid and animal species from which the ADME data was generated needs to be listed.  It looks like this is a mix of data including cypermethrin, fluvalerate, deltamethrin, etc, and human data.  Also, birds don’t produce urine.

P2, L92-93:  Some typographic errors.  Can be added to the previous parasite.

P2, L94:  Remove “however.”

P3, L98-100:  Grammatically awkward.  Maybe make the species plural:  “in rats” and “in mice” etc

P3, L94-106:  The LD data is a mix of different units (mg/kg, g/kg) which is a little confusing to read through.  I’d recommend deciding on a standard unit (usually mg/kg) and being consistent. Additionally, some of this this information could be tabulated to enhance clarity.

P3, L115-117:  Is there any toxicity data on geraniol in quail or in mixtures (with permethrin and piperonyl butoxide). 

M&M: 

P3, L124:  Were these animals housed indoors or outdoors?  Was the grid-bench floor raised?  What was the water source?

Results:

P5, Table 2:  the reference value is from a surviving bird?  You said all 7 died?  Do you mean the longest surviving bird?  If it was a survivor, shouldn’t it say “Affected” and “Survivor” in the top row after “Parameter”? Why was a control bird not used to generate the reference?  Under “total bile acid” did you use two different birds, or was the comma supposed to be a decimal point?

P6, L195:  First letter of paragraph missing.

P7, L215:  solitary cell degeneration?

Discussion:  I’m not convinced that you can blame the lesions on the permethrin when there were multiple other ingredients in the product, and the clinical signs and lesions were not typical of pyrethroids.  Indeed, the lesions were what I would expect more from a terpenoid based on my experience. 

P8, L249:  You need to reference the statement “Based on the scientific literature.”

P8, L261-262:  “the total bile acid levels of liver cell specific values” I don’t really understand this sentence.  Do you mean that total bile acid concentrations are a specific measure of liver cell injury?

P8, L266:  An increase in the potassium concentration is, by definition, hyperkalemia. 

Conclusions:  I think it’s overreaching to state that pyrethroids in general or permethrin specifically were responsible for the morbidity and mortality noted, since the pesticide contained multiple active ingredients and, presumably, inactive ingredients. 

P8, L288:  It was apparently permethrin powder, not pyrethrin powder, and that was only one of several active ingredients.

Comments on the Quality of English Language

There are a few places where I think the English could be improved, as pointed out above.  There were also some typographical errors.

Round 2

Reviewer 3 Report

Comments and Suggestions for Authors

Dear authors

Thank you for accepting most of the previous suggestions, changes and edits. However, there are minor and major editing still pending. The next points should be analyzed carefully, as follows

-. L128 Please use "mountain quails (Oreortyx pictus)" instead of "birds".

-. L135 Please delete the scientific name of the studied birds.

-. L174-L177 This molecular study still needs to be supported by cited references and should be included here. Details referred to the PCR detection of "aviadenovirus", "avian hepatitis virus", and "circovirus (beak and feather disease)" are necessary. Several questions should be addressed. Are the authors analyzing aviadenovirus genotypes related to inclusion body hepatitis? When the authors mentioned "avian hepatitis virus", is this referred to avian hepatitis E? Is this named circovirus the one potentially related to psittacine beak and feather disease? and Why the authors targeted latter viral agent in non-psittacine hosts such as the studied mountain quails?

-. L214 to L234 The histopathologic description is under discussion. The authors highlighted the presence of "Councilman bodies" in this liver. From my view, there are several enlarged nuclei with marginated chromatin containing compatible intranuclear basophilic inclusion bodies together with hepatocellular necrosis, and mild inflammatory cell infiltration. This microscopic finding could be a suspected diagnosis of fowl adenovirus infection, mimicking the microscopic lesions noted in broiler chickens affected by inclusion body hepatitis. A PCR test for fowl adenovirus-inclusion body hepatitis is necessary to confirm this etiology.

-. Based on these comments, I recommend the authors to include a suspected infection due to fowl adenovirus mimicking "inclusion body hepatitis" noted in broiler chicken worldwide, together with the suspected permethrin bath poisoning in the studied mountain quails already identified. 

Reviewer 4 Report

Comments and Suggestions for Authors

I appreciate that the authors were more careful about the use of the terms pyrethrin, permethrin, and pyrethroids this time, though there was still a tendency in the text to generalize ADME and other characteristics between chemicals and species.   For example, geraniol toxicity data from rats cannot be applied to birds.   There is still a strong tendency for over-interpretation of the findings.  This doesn’t appear to be a controlled experiment, and it seems like there was really no control group and many variables that could have played a role. 

Permethrin poisoning in quail

Simple summary:  still using pyrethrin when you mean permethrin.

Introduction: 

P2, L65:  closed parenthesis after molecule with no preceding open parenthesis.

P2, L79:  “Though there is limited pharmacokinetic information on permethrin in birds, the pharmacokinetic profile of pyrethroids (including e.g. . .) is summarized based on the current literature.”

P2, L86:  I think this should be pyrethroids, not permethrin.  If the reference isn’t specifically about permethrin, you can’t use the term permethrin.

P2, L89-90:  Are these percentages based on the total dose?  So 90% of the dose given is eliminated in the urine of mammals?  Needs explanation.

P2, L94-95:  “more sensitive species; felines and poikilotherms are particularly susceptible.”  Add sentence on L96-97 to this paragraph.

P3, L103-104:  “The permethrin is less toxic dermally resulted in higher subcutaneous LD50 . . .”  There are some grammatical errors in this sentence, but I’m really not sure what you’re talking about.  Dermally means topically applied.  Subcutaneously means injected in to the subcutis.  You need to specify if the doses given were applied to the surface of the skin or intradermally.

P3, L112:  “The antiparasitic product used to treat a flock of mountain quail in this study contained geraniol”

P3, L119-120: “The toxic potential of geraniol has been examined in various animal species, including carcinogenicity and mutagenicity tests.”

P3, L125: so we really have no idea how geraniol alone or in combination with other chemicals would impact birds.

Methods: Do we know the inactive ingredients in the pesticide product?  Were the unaffected birds exposed to the pesticide?  Were they treated similarly to the experimental animals, as in were they recently moved?  Are they in similar outdoor enclosures?

Results: 

Clinical Signs:  I am still not clear on this timeline.  It says that the pesticide was removed when clinical signs appeared, but later it says that clinical signs appeared on day 3.  Does this mean that they were exposed to the pesticide for 3 days?  And if they died 4 days after treatment, is that the start or the end of treatment?

Hematology: If these values are from the one longest surviving bird, what are you using for the reference value?

Necropsy:

P6, L193:  Technically, birds don’t have an abdominal cavity because there is no diaphragm.  This would be coelomitis.

Figure 2B:  I’m not a pathologist, so excuse my ignorance, but why is there so much clear space in this slide?  Even beyond the staining, it doesn’t look much like the liver parenchyma in A.

P7, L135-136: “The tubular structure of the kidney was segmentally,”  segmentally what?  “the entire epithelial cell population disintegrated and died, and the tubule cavity was filled with cell debris”  The renal tubular epithelium was necrotic and often sloughed into the lumen?

Discussion:

P7, L244-246:  I don’t understand this sentence.  Do you mean “malfunction” of the liver based on the enzyme changes?  What are “regressive lesions”?  Does “soliter” mean solitary?  What do you mean by “disturbance of fat circulation?”

Discussion: 

P8, L269-271:  We don’t know that the other birds were unaffected, since there was no serum chemistry or postmortem examination of them.  We know that these seven birds were separated from the others and put in an outdoor enclosure, which is likely a source of stress.  Could environmental stress have caused the clinical signs and deaths in a physiologically stressed group of birds?  There was no real control group, as near as I can tell.  Were there other birds moved without exposure to the pesticide?  Were there other birds exposed to the pesticide and not moved?  This is clearly more of a case study than an experimental investigation.

P8, L279:  Based on L125, there is no data on geraniol in birds, so we don’t know if it could have caused or contributed to the clinical signs.  I don’t think that the conclusions that follow that it was definitely permethrin are overreach. The permethrin could have contributed, but you really haven’t ruled out other potential contributing factors.

Comments on the Quality of English Language

There are several grammatical and English errors that were missed from the previous version.
